# Precision Livestock Farming: What Does It Contain and What Are the Perspectives?

**DOI:** 10.3390/ani13050779

**Published:** 2023-02-21

**Authors:** Joachim Lübbo Kleen, Raphaël Guatteo

**Affiliations:** 1CowConsult, Coldinnerstr. 65, D-26532 Grossheide, Germany; 2Oniris, INRAE BIOEPAR, 101 Route de Gachet, F-44300 Nantes, France

**Keywords:** dairy cattle, sensor, health, welfare, veterinarian, precision livestock farming

## Abstract

**Simple Summary:**

Digital tools are becoming increasingly important in livestock farming. The impact of digitalization appears to be best documented in dairy cattle husbandry. Precision Livestock Farming (PLF) is a term that encompasses sensors for the capture of biological information, algorithms processing the information and interfaces that allow for making use of these data. PLF is expected to optimize animal production, health and welfare. PLF provides an enormous amount of animal information that can be used in various ways. While some systems, e.g., estrus detection systems, are already extensively used, other systems await practical application. PLF may allow for progress in animal health, production, and welfare. However, so far there is little scientific evidence that application of PLF does actually have an effect. It is to be expected that PLF will alter the animal–human relationship and will have a substantial impact on veterinary practice.

**Abstract:**

Precision Livestock Farming (PLF) describes the combined use of sensor technology, the related algorithms, interfaces, and applications in animal husbandry. PLF technology is used in all animal production systems and most extensively described in dairy farming. PLF is developing rapidly and is moving beyond health alarms towards an integrated decision-making system. It includes animal sensor and production data but also external data. Various applications have been proposed or are available commercially, only a part of which has been evaluated scientifically; the actual impact on animal health, production and welfare therefore remains largely unknown. Although some technology has been widely implemented (e.g., estrus detection and calving detection), other systems are adopted more slowly. PLF offers opportunities for the dairy sector through early disease detection, capturing animal-related information more objectively and consistently, predicting risks for animal health and welfare, increasing the efficiency of animal production and objectively determining animal affective states. Risks of increasing PLF usage include the dependency on the technology, changes in the human–animal relationship and changes in the public perception of dairy farming. Veterinarians will be highly affected by PLF in their professional life; they nevertheless must adapt to this and play an active role in further development of technology.

## 1. Introduction

Digitalization, the Internet of Things (IoT), Big Data—Buzzwords which are increasingly associated with dairy farming. Not only agricultural and veterinary publications are embracing this topic, the general media also cover it: while a headline like “The Connected Cow: Optimizing Dairy Cow Health And Productivity With Technology” [1] appears to make huge promises, another reads “Big Brother in the Cowshed” [2] and may be understood as displaying a certain caution towards technical monitoring of animals and possibly a deterioration of the human–animal relationship. It is unclear how widespread the adoption of digital technology already is. For Germany, the professional association Bitkom stated in 2020 that 80% of farmers are using some kind of digital tool in animal husbandry [3]. The market for precision farming, including all sectors, is expected to grow at an annual rate of 13% to reach a volume of USD 12.9 billion in 2027 [4]. It is therefore to be expected that digitalization is becoming ever more relevant for all aspects of dairy farming, including veterinary services. This paper tries to give an overview of digitalization in the dairy cattle sector and to exemplify the current technology. It will also give a critical assessment of the impact of new technology on the veterinary profession and the animal–human relationship. It aims to provide the reader with an overview of the state-of-the-art components of PLF, the principles of its functioning and a comprehensive analysis of perspectives, limitations, opportunities and risks of PLF.

## 2. Subject Definition

Digitalization refers to various trends and innovations; it has been proposed that it refers to “changing business models” and “new value producing opportunities” by using digital tools. It has to be differentiated from the term “digitization”, which only describes putting analogue processes into a digital format [5]. In production animal husbandry, the term digitalization often refers to sensor technology, electronic data processing or automated systems, e.g., in milking. Increasingly, the term “Precision Livestock Farming” (PLF; [6]) is being used. PLF includes robots or sensors collecting and producing data which are computed and analyzed by standardized operations (algorithms) to produce relevant information. The algorithms analyze for points of interest (e.g., “alarms”) that serve as support or basis for decision making, mainly by the farmer himself. The main difference compared with traditional, more retrospective decision making is (1) the immediate availability and processing of data; (2) the integration of data from various different sources; and (3) the rapid decision-making process, implementing changes instantly [7,8]. PLF therefore implies a system with components sensors, algorithms and interfaces for making practical use of data from livestock farming; the decision-making process, either automatic or supported by applications, is an integral part of PLF itself (Figure 1).

In animal production, the implementation of PLF seems best documented in dairy farming [9,10]. The development of PLF is fast; in 2008, Wathes et al. [11] referred to PLF as being an “embryonic technology”, only measuring certain parameters in animals. Twelve years later, in 2020, Cabrera et al. [12] presented the “Dairy Brain”, an integrated system automatically optimizing group feeding and providing early recognition of cases of clinical mastitis. Thus, PLF appears to be developing away from only measuring parameters towards integration of different components and decision support or itself making decisions [13,14].

## 3. Elements of PLF

### 3.1. Sensors

A “sensor” is commonly defined as “a device that is used to record that something is present or that there are changes in something” [15]. There are numerous articles reviewing available sensor technology, reflecting the current state of knowledge at the time of publication [16,17,18,19,20]. 

In dairy cattle husbandry, “sensor” still seems to be associated primarily with automated heat detection based on activity data. This application is widely being used and is well researched; its practicability had already been documented more than a decade ago [21]. Kempf [22] reported a regular estrus detection rate of 95% using activity monitoring for estrus detection and generally higher effectivity compared to visual heat detection. This study is just one of many examples for estrus detection based on activity. The use of these sensors seems to be economically advantageous [23] and may in the future be integrated with physiological data to target cows more individually for even higher reproductive performance [24]. 

Most parameters around an individual dairy cow or the herd can be monitored by sensors. One biological parameter may be measured by different sensor types. Rumination monitoring may serve as an example: it may be captured using accelerometers [25], pressure sensors at the head [26] or microphones [27]. Scheurwater et al. [28] showed the possibility to assess rumination or other activities using an intrareticular pressure sensor. The device was able to identify rumination with an accuracy of 0.92, a sensitivity of 0.97 and a specificity of 0.90. 

Various sensors are currently available and regularly connected to data systems—they can be understood as the “Internet of Things” (IoT) application to animal husbandry (Figure 2). Knight [15] categorized sensors according to their relative position to the animal into “at cow” sensors (e.g., accelerometers and ruminal pH sensors), “near cow” sensors (e.g., cameras or microphones) and “from cow” sensors (e.g., real-time in-line analytic systems for milk; sensors covering “not-cow data” such as room thermometers also have to be included here). The author further states that while a lot of systems are commercially available, they may not all have been assessed for their viability and validity in a scientifically sound manner. Inversely, various technical options have scientifically proven potential but are not commercially available. Rutten et al. [14] reviewed the development of sensor usage and the changes in their practical application. They describe sensors developing away from being isolated tools, instead becoming integrated into a more complex system on the farm (Figure 3). The phases described are (1) the sensor as tool measuring a biological or physical parameter and delivering this information; (2) interpretation of this data, e.g., an estrus alarm on the basis of changed activity or a ruminal fermentation disorder on the basis of intraruminal pH value patterns; this “alarm function” rests on a standard derived from already retrieved data, the validation of sensor data and finally an algorithm that translates measurements into an alarm. The authors describe as next step (3) data integration, combining various sensor and other information. An example would be the combination of activity and rumination data with farm data such as reproduction information or health events. Those data are generally provided externally by farmers themselves, veterinarians or milk-recording organizations. This enrichment of data leads to a more precise identification of relevant events, e.g., estrus or disease. The algorithms (see below) achieving this identification are trained to recognize data patterns, make predictions on the basis of these patterns and forecast relevant events. Animals at risk of a certain disease can thus be identified early and receive special attention [29,30]. It is not only disease that can be predicted; another example would be the expected reproductive outcome of an individual animal or the herd as a whole [31]. While this information is so far used only for decision support, the next step according to Rutten et al. [12] would be (4) decision making. PLF information is so far mainly being used to make qualified decisions. Decision making by PLF components would suggest decisions to farmers; it would “provide fast, concrete and simple answers to complex farmers’ questions” [32]. Decision making applies to various aspects of herd management; most relevant today are decisions in the field of reproduction and animal health (“alarms”) [33]. Herd dynamics and replacement (culling), which appear difficult to model, will certainly be aided by the information provided in a PLF environment [34]. An integrated system can therefore serve as a central hub to collect information from various sources and provide informed decisions covering all areas of herd management including feeding, culling, treatment options and alike [12]. This appears to be the most inclusive development of PLF in a true sense. Sensors are mere providers of information, which are linked with other data from various sources. It has to be stated that so far there is no commercially available system that offers a “complete” monitoring of herds or individuals. Systems tend to be “stand alone” solutions, monitoring only one or two parameters; moreover, sensor systems are rarely communicating with each other, thus limiting the possible insight. As a consequence, farmers will have to rely on a combination of different PLF components if they want to realize full monitoring of their herd [17]. It is therefore important for the whole industry to promote collaboration and communication within PLF [20].

Sensors largely represent the “hardware” component of PLF and often closely interact with the animal. As they are at least partly used for medical purposes, they would, if used in humans, have the status of medical devices. These require specific regulations to ensure health and safety. As surprising as it may seem, there is hardly any regulatory framework concerning sensors in terms of both expected efficacy and safety. Some sensors are attached directly to the animal or even inserted into it, e.g., the digestive system. The disposal of sensors still present after slaughter and the potential of endangering food safety after culling may be seen as risks for human health and the environment. Alarms produced by PLF components may lead to some form of treatment or even culling of an animal; hence, there is a certain risk in system malfunctioning or misinterpretation leading to issues in animal health and welfare. The implementation of a harmonized methodology within one market (e.g., the EU) or across markets (e.g., by an international body) for evaluating these tools and a material vigilance system would undoubtedly be necessary. A licensing system, approval or guarantee scheme safeguarded independently appears to be desirable. 

### 3.2. Algorithms

An “algorithm” is being defined as a “set of mathematical instructions or rules that […] will help to calculate an answer to a problem” [35]. It is therefore a standardized, iterative process which produces information into “specs” needed for decision making.

As previously mentioned, algorithms are at the core of PLF as they create meaningful and applicable information out of the pool of available sensor data. They may include data from other sources, e.g., milk recording. The algorithms applied have to cope with numerous challenges: (1) While the current amount of data produced by most systems is generally moderate in size [17], an increase is to be expected; for example, the more common use of photography-based sensors could lead to huge volumes of information. (2) If systems are interconnected, the data produced is multi-dimensional in nature, including rather simple information, e.g., animal ID, but also physical measurements, optical information and data entries from outside sources. (3) It is also necessary to account for missing, wrong or redundant data, often repetitive in nature or only slightly changing over time. Nevertheless, this less-than-perfect information needs to be processed and relevant patterns recognized. (4) In order to be useful and allow for proactive decision making, the algorithms generally need to involve a prediction part that identified events of interest likely to happen in the future [36]. These challenges can be mastered by application of “machine learning” in which algorithms are trained to recognize relevant parts in a pool of information. This has been extensively reviewed elsewhere [37]. Development of an algorithm can occur in two ways, depending on whether or not the point of interest is known: (1) Information or events that are already known can be analyzed for their informational characteristics and used by an algorithm to be recognized in another data set. (2) The algorithm itself identifies relevant characteristics in sensor data and defines the event; the algorithm thus identifies conspicuous patterns in the data that can retrospectively be linked to an event of interest. This process can therefore be defined as either “supervised” or “unsupervised” learning [10]. Dittrich et al. [38] describe the structure of algorithms for recognizing events. Numerous clinical studies have identified measurable behavioral changes in animals with certain diseases. For example, cows with clinical mastitis show different patterns in lying and standing times, and their patterns of activity and feeding behavior change. These changes are captured by sensors and will be analyzed by algorithms using the recorded patterns of healthy animals as a standard. Deviations from this standard point to the occurrence of clinical mastitis. As behavioral changes usually occur prior to the onset of clinical symptoms, the algorithm does not detect the clinical disease itself but rather predicts its occurrence.

Reiter et al. [25] describe the validation of an algorithm belonging to an accelerometer, giving an example of supervised learning. The sensor is attached to the ears of cows and measures motions in a three-dimensional way. The study does not validate the accuracy of the sensor itself in capturing motions; it rather shows the approach to train and validate the ability of the corresponding algorithm to recognize rumination in the motion data set produced by the sensor. The sensor was attached to ten cows, and observers visually identified time segments with rumination. The time segments were then marked in the data set and processed into the algorithm. In a second step, the thus trained algorithm was tested for repeatability of motion pattern recognition in a new data set by comparing sensor labels with visual observation. The study shows a highly correlated agreement between visual observation and recognition by the sensor. For example, the sensor identified an average of 1508 s spent ruminating per hour of resting phases; observation by a human recorded 1492 s of rumination, a mere difference of 16 s per hour spent by the animal resting. 

It is usually not known how exactly an algorithm in a commercially available system is built. The algorithms are typically not accessible in publications as this is commercially used intellectual property and therefore sensitive information. Some studies, however, illustrate the development of an algorithm. An example is the study by Carslake et al. [39], demonstrating the practical implementation of algorithms on the example of the behavior of calves. The authors describe the “training” of a sensor from mere measuring activity up to the interpretation of captured data. In this study, the activity and behavior of 13 calves were captured by an activity sensor and at the same time recorded on video. The aim was to correlate both sensor and visual information. The incoherent raw sensor information on activity was therefore “mapped” with labels derived from the visual information captured on video. As a result, it was possible to identify activities such as licking, suckling or playing in the sensor data set. This would allow for recognition of these activities in the field; changes in the behavior could then be identified, this possibly indicating disease. 

### 3.3. Applications

Sensors and their respective algorithms are an important element of PLF. These components produce data, either raw or processed, and this data has to be made accessible and meaningful. Applications achieving this are often computer systems installed on the farm; they form a database that contains farm-related information, e.g., production and fertility data or milk production. They can also serve as a platform for sensor-related applications and directly provide communication between these components, serving as an interface between farm production data and sensor-related information. Herd management programs, originally designed for storage and analysis of farm production data, are increasingly serving as exchange platforms combining data originating from the farm. This is developing into a two-way process because alarms from a sensor system are useful to be recorded and, inversely, a sensor-related application will benefit from farm records such as insemination, calving dates and alike. In recent years, a shift towards cloud-based solutions is occurring, in which farm data including sensor information is transmitted towards an external data storage and processed centrally [40]. Relevant information is available to the farm by specially designed applications, through websites or via alarms, e.g., on a mobile phone [41]. There is hardly any scientific literature on the nature and effect of these applications. Rarely, the analyses of herd management programs are used for research, an example being the study by Sorge et al. [42], in which the economic evaluation of dairy cows in the DairyComp^®^ program suite was thematized. In any case, the amount of available data on and from dairy farms appears to be increasing, so too does the need to apply this information into practical decision making. It appears useful to make the generated information simple to grasp by proper design, e.g., by means of graphic illustration [43].

### 3.4. Interfaces

Connecting data originating from different sources needs interfaces; they link electronic farm-specific data (e.g., insemination dates) with external data (e.g., milk recording data) and data from sensor systems, which may originate from different manufacturers. However, there is a tendency towards closed “ecosystems”, in which components from a single manufacturer are combined. The exchange of information between components of different manufacturers may be difficult to establish. Bypassing the ecosystem barrier can be achieved externally by a so-called “warehouse”. Relevant data from various sources on one farm is uploaded onto a server; there, it is collected, cleaned and sorted. Relevant information for certain analyses is then extracted from the warehouse and transformed to answer the questions regarding an individual or a group of animals, an individual farm or a cluster of farms [44]. In the long term, linking farm-specific data with data from other sources will facilitate automatic decision making at the farm level. Ferris et al. [45] introduced an integrated system which not only uses all available information at the farm level but also information such as weather data and pricing information. A system like this is no longer designed to measure processes or raise alarms; it predicts production-related processes at the herd level and proposes management decisions or implements these decisions autonomously. Cabrera et al. [12] illustrate the use of such a system on the example of udder health. Animals can be identified as being at high risk for developing mastitis as early as in their first lactation. This classification is achieved by using genetic information [46], animal behavior and production data from the milking parlor or milk recording. The system gives recommendations for this animal such as exclusion from breeding or early culling due to low potential. 

Another concept integrating data is the “digital twin” [47], in which available information is used to create a virtual image, a doubling of a certain entity. In the case of dairy farming this could mean the ‘twinning’ of an individual animal, a group of animals or a herd as whole. The digital twin can then serve as a benchmark which points to risks in the future as it anticipates certain developments. This may also help in objectively assess animal welfare. The principle of this is already used in teaching so that the effect of certain decisions on a dairy herd can be modeled and studied [48].

## 4. PLF Influencing Animal Husbandry

The rapid development of PLF and the possibilities it creates are evaluated differently in the literature. There are few evidence-based studies demonstrating a measurable effect of PLF usage on animal health or productivity. Knight [15] states that while sensor technology can measure a lot of biological parameters and has become very effective in recognizing disease or disease risk, it remains unknown what the effect of their presence on the production and health parameters really is. Determining this effect faces the problem that this can only be done indirectly. As long as a PLF system on a farm is not acting and deciding fully automatically and autonomously, it may only influence human behavior. Individual farmers are likely to respond differently to automatically generated information, and their decisions may be affected by PLF to a varying degree. Eckelkamp and Bewley [49] evaluated the behavior of dairy farmers using PLF technology. The study assessed the intensity with which farmers evaluated the health alarms generated by PLF systems and how they responded to this information. It was shown that farmers generally considered the alarms to be “true”; this did, however, not automatically trigger a certain behavioral response, e.g., examining an animal. Especially if farmers themselves did not expect animals to have problems (e.g., cows not in the transition phase), alarms tended to be ignored. The number of alarms generated by the system affected the probability of farmers acting: at 20 or fewer alarms per day, this probability was higher. Furthermore, the authors showed a “habituation”: the longer the system was present on the farm, the more likely alarms were to be disregarded. This points to a risk of PLF application: the presence of PLF systems is no reliable indicator for a better or improved situation concerning animal health and welfare on a dairy farm. As PLF systems continuously produce data (and sometimes alerts in the case of deviation of observed data from expected data), stress can be put on the user, especially when the tools lack specificity (alerts generated on non-diseased animals). Therefore, a difficult compromise needs to be made between sensitivity to detect alterations and acceptable specificity, as a specificity that is too high will lead to false positive alerts for the user. This is illustrated by a study focusing on the prediction accuracy of a system monitoring rumination and activity; the sensitivity was programed to produce less than four false positive alerts/100 cows/day. As a result, however, only 30% of disease events were detected, based on rumination and activity drops [50].

Faced with this incessant flow of data and alerts, the owner may lose confidence in the tool and even may altogether stop using information at the risk of missing sick animals. There is also a risk that alarms may be “over-used” with no critical review or examination of the animal. The possible advantages of PLF usage therefore largely rely on the performance of the tools and their operating conditions. These behavioral aspects of PLF application need to be considered in development of future applications. 

In summary, the effect of PLF applied on dairy farms remains unclear. In a study on Italian dairy farms [51], no higher milk production was found on farms using sensor technology. A Dutch study [52] evaluated the effect of sensors being used for udder health and fertility monitoring on related parameters. Only small effects were found, and milk production was not better on farms using the sensor systems. The sensors at that time were, however, able to generate only very few and selective health alarms. Various sensor-based systems have been proposed for detection of lameness in cattle; however, their usefulness and cost–benefit ratio are often not acknowledged by farmers despite lameness being a major animal health issue [53]. It therefore appears necessary to demonstrate the benefit of this technology under farm conditions. As mentioned before, a scheme to objectively assess the impact and risks of a PLF system appears to be necessary.

## 5. Animal Welfare and Animal Health

Animal welfare concerns numerous aspects going well beyond animal health alone; while health is important, animal welfare also involves aspects of animal behavior and emotion [54]. Sensor technology can capture various aspects: behavior of the animal (movement and activity), time budget, but also vocalization and social interaction can be monitored. Furthermore, physical parameters affecting behavior, e.g., ambient temperature, are monitored easily. Combining this information may very well contribute to a general improvement in animal welfare on dairy farms [55]. Due to the constant nature of the systems, PLF may be able to monitor animals in a more intensive, more reliable and more consistent manner than traditional animal monitoring by humans alone is able to achieve [17]. Some examples below illustrate the possibilities PLF is opening for the management of dairy farms with special respect to animal health and welfare. This list can by no means be exhaustive and covers both technology that is already available and also possibilities that so far are merely a concept. 

The previously mentioned monitoring of rumination is possible using different technological approaches and may serve in the early detection of disease, especially metabolic conditions. In a study using 312 dairy cows, Gusterer et al. [56] showed that changes in rumination activity detected by a sensor system markedly preceded clinical diagnosis of a disease. The study used the monitoring of rumination frequency and length of rumination bouts and compared this information with the results of daily standardized clinical examination with ß-hydroxybutyrate measuring, all in the period from calving until day 8 post-partum. It was shown that rumination monitoring identified clinical disease up to five days before a clinical diagnosis could be made. 

Optical systems to determine the body condition score (BCS) of cattle have become commercially available in recent years. The BCS is generally understood as the main risk factor for metabolic disease in the transition period of dairy cows [57] and is mainly monitored using traditional adspection methods. Optical monitoring using a three-dimensional capture of body points is more sensitive to changes and describes the BCS better and more consistently than the traditional system using half or quarter scoring points [58,59]. As the BCS is a parameter intricately linked to metabolic processes, these systems open the door for high-throughput phenotyping to finally improve genetic selection on resilience parameters.

Monitoring locomotion health in dairy cattle may benefit from PLF technologies. Digital systems that are collecting and sorting diagnoses from hoof care have been available for some years; they are helpful to give a more systematic, dynamic and more objective picture of diseases of the feet. [60]. In particular, control of digital dermatitis may be improved by digital documentation as the associated algorithms help to describe the development on herd level better and aid in identification of herd-specific risk factors [61]. Animals experiencing lameness show a different locomotive behavior [62], and the transition from soundly walking to mildly lame is regularly missed by farmers, the identification of which would help in controlling lameness at the herd level [63]. The use of accelerometers has been described for lameness detection and appears to be a reliable method [64,65]. Early detection of locomotion disease is also possible by use of optical monitoring systems, which, similarly to the abovementioned principle of metabolic disease detection, may recognize clinical alterations earlier than traditional methods and thus help to improve locomotion health in a herd [66].

Heat stress is regularly identified as a challenge to animal wellbeing and production. So far, most farms operate their cooling regime on fixed schedules or according to ambient temperature. The actual level of heat stress can be predicted using environmental and animal production data and hence provide a more exact description of the challenge [67]. The inclusion of actual body temperature data may refine strategies, making heat abatement more effective [68]

The control of feeding dairy cows is regularly impaired by the challenge to correctly assess the actual dry matter intake of the animals. Optical monitoring using the technique of “photogrammetry” may provide a more reliable and constant information flow. Photogrammetry converts optical data, i.e., the volume of feed, into mass. The monitoring of changes in volume and consequently in mass allows for the calculation of feed intake by the animals [69]. Generally, the connection of various technical components related to the feeding process offers the perspective to make feeding more animal-oriented and efficient [40].

Capture and analysis of livestock vocalization is so far merely a concept. It has been shown, however, that different animal vocalizations may be sorted and classified [70]. In the long term, this may help to assess animal affective states in a quantitative and thus measurable way, allowing for an objective evaluation of animal welfare in dairy herds. Thus, the assessment of biological data from various sources may enable the analysis of “affective states” [54], which so far form a rather inaccessible part of animal welfare [71].

**Figure 1 animals-13-00779-f001:**
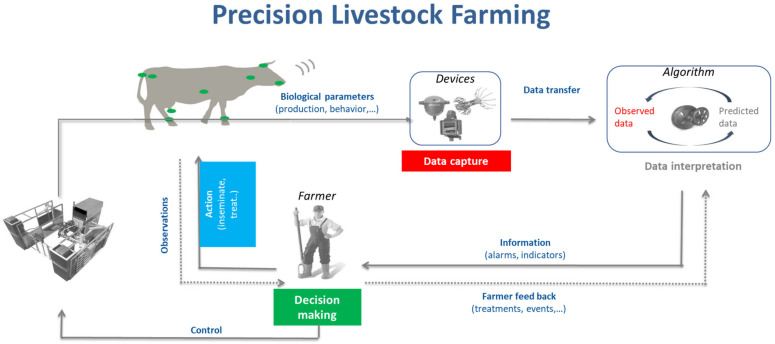
Overview of a PLF system of various components on a dairy farm. The automated milking system (AMS) in this case serves as a complex of sensors capturing data from an individual animal. The attached algorithm interprets the data and creates predictions that are transmitted to the farmer to aid decision making [71].

**Figure 2 animals-13-00779-f002:**
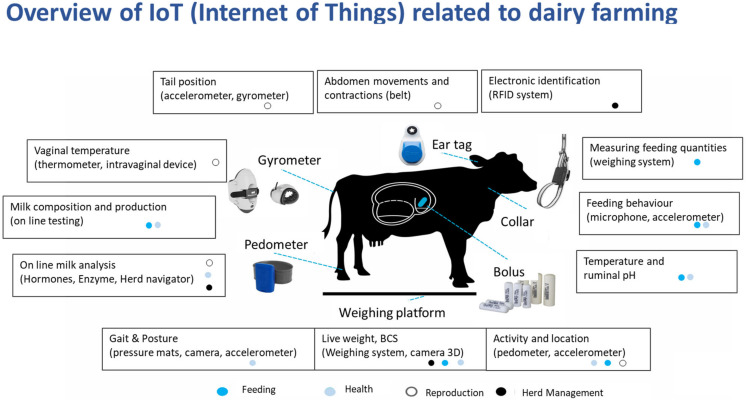
Overview of currently used devices to capture biological data from animals. All these sensors are connected and can hence be understood as the application of the “Internet of Things” to the dairy cattle sector.

**Figure 3 animals-13-00779-f003:**
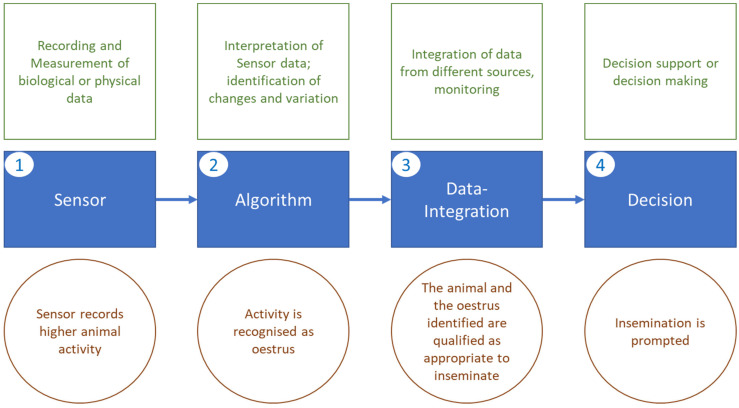
Elements of Precision Livestock Farming, adapted from Rutten et al. [14]. While sensors are measuring parameters, algorithms are recognizing relevant changes and interpreting them (here with the example of estrus detection). Additional information from other sources helps to classify the event. The final stage is a system that autonomously acts on the basis of the information—in this example, it decides on inseminating the animal.

With respect to animal breeding, the large-scale and continuous collection of phenotypic data (e.g., milk production, growth rate, behaviors and disease resistance for example) opens the way to what is known as “high-throughput phenotyping”, i.e., the characterization of all the apparent characteristics of an individual. This can happen continuously and almost in real time using connected sensors and tools. This phenotyping allows for genomic studies that rapidly select animals carrying the characteristics of interest (such as disease resistance, a phenotype that is very difficult to characterize classically). It may also open up perspectives for phenotyping on traits not currently available and hence creating new possibilities for selection. To do this, it is fundamental to associate different people from different backgrounds to develop the tools, not forgetting to associate the end user in particular [72]. Connected tools could make it possible to bring out the individual in the group and thus give visibility for the breeder to isolated individuals, especially in large numbers. However, the opposite effect is also realistic; an extreme standardization of the animals could lead to genetic impoverishment by eliminating individuals that go beyond the hoped or expected standards.

## 6. Human–Animal Relationship

The increasing presence of PLF technology on dairy farms is likely to change the interaction between man and animal. An “alienation” further separating the animals from the people responsible for them appears possible. The animal would become a mere functional component in an increasingly efficient production system. Inversely, PLF seems to offer huge potential to better recognize and respond to animals’ needs and prevent factors negatively influencing their wellbeing. This spectrum of possibilities is laid down in a report by a Dutch study that gives a comprehensive overview on the respective risks and opportunities [73] (Figure 4). Covering the different perspectives of the animal, its owner and society in general, the study tries to gauge positive or negative consequences digitalization, electronic monitoring and automated decision making might bring to animal husbandry. Opportunities are generally seen in a more exact provision and care for the animals’ needs, an early detection and prevention of disease; an incorrect usage of the systems and the information received from them or malfunctioning of systems themselves are identified as major risks. The authors recognize the danger of reducing the animal to a mere function and “thing”, i.e., that animals may be understood as a pure functional part of the production process, discarding its properties a sentient being. From the producer’s perspective, PLF opens perspectives to a more efficient production process and reduction of workload; on the other hand, the digitalization makes a farm vulnerable to data theft or cyber-attacks. Moreover, a certain dependency on the manufacturers of the systems is created. From the perspective of society, there is the chance to create more transparency of the production process, as animal husbandry may better be understood, and its principles more effectively communicated. However, extensive application of PLF technology may alter the image of animal husbandry towards being “over-engineered”, this conflicting with the traditional perception of the farming sector. This conception may also be shared by farmers themselves. However, individual preferences are to be considered. For example, PLF-associated technology can be perceived merely as a tool not changing the production system itself at all. Interestingly, farmers that have a strong emotional relationship with their animals seem to recognize the potential of technology more positively [74]. An ambivalent attitude in society is reported by Krampe et al. [75]. The study evaluated the perception of PLF in consumers of three European countries; the already mentioned “over-engineering” of animal husbandry was identified as a concern. However, opportunities were recognized by consumers nonetheless: improvement of animal health and welfare, a positive impact on sustainability and transparency of the market-chain were mentioned. The information provided by PLF technology can very much alter the perception that the breeder or owner has of his animal. PLF allows for a farmer to go beyond the basic, production-related information of his animals: now, movement, behavior (feeding, sleeping) and exact location can be monitored and assessed. This may allow for a better understanding of the animal and a more personalized approach, bringing back the “individual animals perspective” even to very large herds. 

The absence of a specific regulatory framework for hardware components on animals means that the choice to equip a group of specific animals with a device is often made by the owner exclusively. While it is obvious that there can be no question of obtaining the direct consent of the animal, it is legitimate to question the circumstances in which one can freely decide whether or not to equip an animal, particularly when the objects are of an invasive character. Expertise from a specialist in animal health or behavior seems useful, especially when validating the equipment and, if necessary, the choice of technical solution. It is also essential that users be properly trained in all the risks implied by the tool.

## 7. Summary—PLF from the Veterinarian’s Perspective

Precision Livestock Farming is a collective term that describes a rapidly expanding complex of sensor technology, algorithms and applications. PLF aims to optimize animal production, health and welfare. It has become impossible to fully describe and evaluate all developments, especially as only a few have been evaluated scientifically. As has been shown on the example of dairy husbandry, the interaction of sensors and information technology creates new possibilities such as early disease recognition, monitoring of animal behavior and possibly improved animal welfare through optimized management. However, neither alarms nor information seem to be the objective of PLF; it is rather an integrated view on the farm, and possibly the whole industry, being interconnected. Decision making, not decision support, is the perspective. The information available increases in accuracy, complexity and volume. The data available may allow for a closer observation of farm events and development, better preparation of clinical activities or advice due to better data at hand and a more effective follow-up after changes have been implemented. 

This creates opportunities but also risks: the dependency on algorithms and their correct function and also the impossibility of comprehending all information collected and created may be hazardous. The veterinarian may be reduced to a component within the decision-making process which he or she may not be able to control or influence. Furthermore, the human–animal relationship may be affected, with the individual animal being viewed as a technical component. The same may apply to veterinary practice with treatments automatically evaluated and suggested beforehand and their success already being predicted based on historic data. This would clearly conflict with the ethical commitment of the veterinary profession. While PLF systems can provide early detection or even prediction of disease, it is unclear how the veterinarian can deal with this. An animal that has been predicted to develop clinical disease can nevertheless still be clinically normal; a treatment using pharmaceuticals is hardly permissible if the indication is not stated by a veterinarian. This touches the fundamentals of the veterinary profession as it challenges the clinical examination and diagnosis. 

The opportunities are there, nevertheless. The systematic collection and evaluation of animal records of all kinds opens new perspectives in herd health. Continuous and possibly remote access by the veterinarian to the data generated by PLF systems opens interesting perspectives in terms of teleconsultation or tele-expertise, creating new perspectives in optimizing health and wellbeing of the animals. This may especially be interesting for farms in remote areas, away from centers of veterinary expertise. However, the help that these systems could bring cannot hide the need to address the issue of planning resources and permanent health monitoring. The mass of data generated may make it possible to rethink the client–veterinary relationship, developing the field of telemonitoring and an “increased” clinical examination for the veterinarian, who would thus have access to data otherwise not available. Conversely, the farmer should not be overwhelmed with information and should contact the veterinarian in a manner that is agreed between both parties. It is indeed the complementarity of the approaches that should benefit the animal. The challenge is then to explain to customers what attitude to adopt when faced with these tools, which cannot entirely replace cow-side care. The use of PLF technologies will also require veterinarians and owners to be adequately trained in the use of these tools and the data and alerts they generate.

Early detection or prediction of disease may very well improve animal health on dairy farms; monitoring lameness in particular could benefit from kinetic or optical sensor technology. This might also help to raise acceptance of the dairy sector in the consumers’ perspective. In any case, PLF is going to alter the working routine of veterinarians. It is not up to the veterinary profession whether to accept these changes—they must be recognized, embraced and used. It is necessary for veterinarians to understand the functioning of PLF and to actively take part in its development.

## Figures and Tables

**Figure 4 animals-13-00779-f004:**
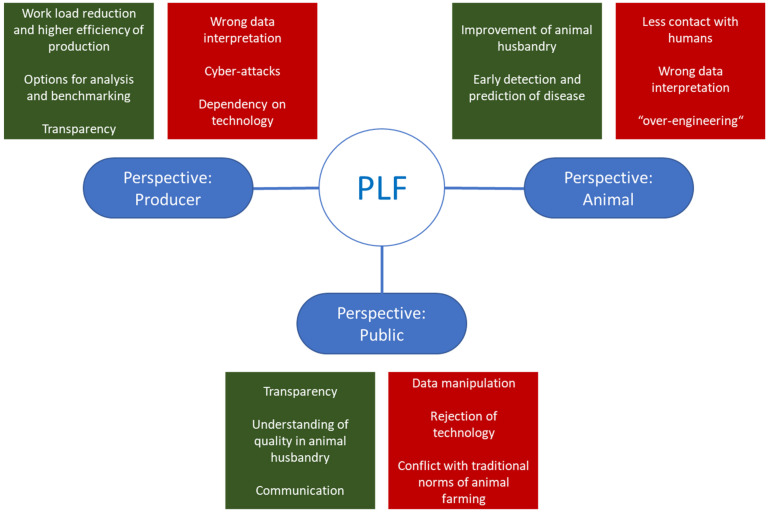
Evaluation of opportunities and risks of PLF. Summary based on (Raad voor Dierenaangelegenheiden [72]). From the different perspectives (animal, producer, public), different options arise.

## Data Availability

Not applicable.

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
