# Peer review of "Precision Livestock Farming: What Does It Contain and What Are the Perspectives?"

_animals, 2023, doi:10.3390/ani13050779_

Round 1

Reviewer 1 Report

The paper is well written, the subject is clear, it is necely structured and the approach is comprehensive, even for a public of non specialists. The literature review is extensive, complete and accurate.

Chapter 4 has a nice discussion of limitations and challenges resulting from the use of PLF in dairy farms. Chapter 5 covers the main dairy cow welfare issues to be takled and nicely discusses positive contributions from PLF to problem moniotoring and control. Chapter 6 and 7 follows previous comments. 

1. Sugestion: review carefuly the use of the word "control", because some times (e.g., line 314) is used instead of monitoring (or analogous word)  

2. "Precision Livestock Farming is a collective term that describes a rapidly expanding complex of sensor technology, algorithms, and applications." (lns 421, 422) there is a lack of purpose in this definition, it would be excelent the inclusion of : what is the general purpose of these aplications, in the definition, before discussion.

3. Not refered in the "references" section: Alain et al, 2016 Fig 1; Livre blanc vet in mentioned in Figure 2;  

4. Comment on Section 7 - Summary

4.1. I would suggest that Fig 4 could be modified and improved by adding the "veterinary perspective" a fourth dimension, inspired by the discussion made on this section.

4.2. As mere suggestion: in my opinion this discussion could be elarged to include the "Herd" dimension, which is a level above the indivudual animal level. Discussion of contributions from PLF to herd health monitorig and fprecasting would be interesting.

Comments regarding review are as notes on the PDF and PPT documents.

Author Response

We would like to thank the reviewers. Their valuable input has greatly contributed to the improvement of this paper.

In the following, we would like to address remarks made by the reviewers concerning content, scientific validity and completeness of the article. Recommendations merely concerning language, wording and grammar have been implemented in the text and are not further discussed here. The manuscript has once more been assessed for readability and changes have been made to improve it.

Reviewer 1:

  1. Definition of PLF: The goals of PLF have been added to make the definition more relevant.

  1. “control” – The use of this term has been revisited and monitoring been inserted where appropriate.

  1. The reference has been added.

  1. 1.: The figure is a summary of the cited Dutch report by the “Raad voor Dierenaangelegenheiden”. As the original report does not include a “veterinary perspective”, it was not included in the figure. We trust, however, that the article does cover this perspective extensively.

4.2.: Although this is a brilliant suggestion, again, the figure relates to an original reference (Dutch report) which does not include this perspective. Especially in section 3.3 (Applications) and section 5 (Animal Welfare & Animal Health) the herd is the major point of interest. We feel that this perspective is therefore covered extensively in our paper.

Other remarks by Reviewer 1:

  • Definition of Digitalization: The definition has been expanded, a reference added and has been made more precise.
  • Health events: In practice, this term is regularly also used for treatments like vaccination or veterinary standard controls. We adapted the term in to “disease events” but refrain from trying a definition as this would add little value to the paper and may rather cause confusion.
  • Definition of PLF in the final section has been expanded to include the goals.

Reviewer 2 Report

This paper presents an overview of Precision Livestock Farming (PLF), a digital approach to livestock farming that involves the use of sensors, algorithms, and interfaces to collect and process biological data from animals.

While the abstract suggests that PLF has the potential to improve animal health, production, and welfare, it also acknowledges that there is little scientific evidence to support this claim.

However, the limitations of PLF usage are not fully explored.

The papar seems to present the following limitations to be addressed:

-In the algorithmic section, both supervised and unsupervised methodologies are listed, but it is not clear how unsupervised algorithms are being applied

-The paper mentions some applications of PLF in detail, including the accuracy of collar applications, but it does not mention other studies that have achieved higher accuracy in detecting ruminating contractions and movements

-The lack of accuracy in certain applications is not addressed in the paper. A summary table with accuracy metrics by application and time would be helpful in understanding the limitations and potential of PLF.

-Finally The aim of the paper is not explicitly stated in the paper.

Author Response

We would like to thank the reviewers. Their valuable input has greatly contributed to the improvement of this paper.

In the following, we would like to address remarks made by the reviewers concerning content, scientific validity and completeness of the article. Recommendations merely concerning language, wording and grammar have been implemented in the text and are not further discussed here. The manuscript has once more been assessed for readability and changes have been made to improve it.

Reviewer 2:

The tasks specifically given by reviewer 2 were:

  • In the algorithmic section, both supervised and unsupervised methodologies are listed, but it is not clear how unsupervised algorithms are being applied.

It is indeed unclear in the original text that one example for each was given. Now the two studies have clearly been marked as “Supervised” and “unsupervised” learning, respectively.

  • The paper mentions some applications of PLF in detail, including the accuracy of collar applications, but it does not mention other studies that have achieved higher accuracy in detecting ruminating contractions and movements.

We have tried to give an overview of various sensor types currently used in Dairy PLF. Collar-attached sensors are in fact nowhere specifically mentioned, however some accelerometers are indeed collar attached. In the section “Sensors” various ways to detect rumination activity have been mentioned, we have added a reference to identification of rumination by assessing intrareticular pressure.

  • The lack of accuracy in certain applications is not addressed in the paper. A summary table with accuracy metrics by application and time would be helpful in understanding the limitations and potential of PLF.

The reviewer is right as there is a deficit in information on accuracy of applications. It is the aim of the paper to give an overview of the PLF as a whole and the implication this development has to the dairy sector. In the ‘sensors’ section the paper tries to be exhaustive in listing the various technical ways to monitor, in the ‘animal health and welfare’ section, it gives examples to illustrate how PLF may change the situation. It is, however, not the aim to give a complete overview of currently marketed systems with their accuracy figures. It is even more difficult as only few systems have undergone a scientifically sound assessment for accuracy. Instead, in this paper there are various references to studies analysing the accuracy or giving this overview for certain areas of PLF, respectively. We feel that a list as suggested by the reviewer does not fit into the scope of this already rather lengthy paper.

  • Finally The aim of the paper is not explicitly stated in the paper.

We have supplemented the introduction with two sentences explicitly stating the aim of the paper. We feel that the conclusions-section covers the aspects.

Reviewer 3 Report

Precision Livestock Farming develops very fast in recent years. While it is still in the initial stage of development and triggers some issues as many of its technologies or applications are not yet mature in practical use. It is essential to know the current objective facts of PLF to clearly recognize its deficiencies and what needs to be improved. This manuscript mainly focus on the elements, practical use concerns of PLF and its influence on society. However, some arguments need to be justified. There are some grammatical error and some sentences are hard to read. Suggest to do carefully check or go through a professional editorial service.

Line 41: Please check the quotes and  “and” in this line, make sure they are correctly used.

Line 53: I think the definition of Digitalization is inexact. As far as I see, “various trends and innovations” is the apparent phenomena or the way of realization rather than the definition of “Digitalization”. It transforms the complex and changeable information into measurable figure or data using sensors and automated processing systems to assist production and management.

Line 55, please check the punctuation, especially the quotes,make sure they are correct. Similarly hereinafter.

Line 66-67: please rewrite this sentence, make it more understandable. Change “production animal husbandry” to “animal production” or “animal husbandry”.

Line 67-69: This sentence is very hard to read. The development is fast, from 2008 to this moment, what progress has been made? Please further explain.

Line 82 and line 100: Start with the author's name. similarly hereinafter.

Line 88: Suggest to change “all parameters” to “most parameters”.

Line 101: “less an isolated tool” or “an less isolated tool”? please check.

Line 102: In the connotation of PLF, sensors monitor not only the biological or physical parameter of animal, but also  parameters affect animal production and daily management, eg. indoor climate, energy consume of equipment, operation of control systems, etc.

Line 117-120: Check this sentence, it is very hard to read.

Line 141:  [] ?

Line 135-137:  Suggest further explain about the expected functional orientation this regulatory framework.

Line 143-160: What are the demands and technological difficulties on algorithms for PLF in practical application scenarios? Machine vision and deep learning are widely adopted in PLF. Except for machine learning, deep learning also needs to be considered. These two different kind of algorithms had different requirements on hardware and data amount.

Line 174:  In this paragraph, the author states that it is usually not known how exactly a model in a commercially available system is built. This makes no sense to me.  I think the key point is to evaluate how accurate the model performed in application test and to improve model accuracy or provide precise models to meet the needs in practical use. This largely depends on the richness and largeness of data during model training to improve its ability to cope with various emergencies. 

Line 266-267:  The significance of alarm in PLF is to remind the potential problems existing in production. The ignore of the alarm is the reaction of the farmers to the alarm based on their experience. The main issue is how to update the system according to knowledge base and make the alarm more precise and reliable. In my opinion, this is not a risk but a demand of improvement in PLF to meet the practical need.

Line 277:  The author concluded that the effect of PLF applied on dairy farms remains unclear. I am not convinced with this argument. The effect of PLF should be comprehensively assessed by both direct and indirect benefit such as labor costs and data benefits. Moreover, the effect of PLF depends on whether it is properly applied. As far as I see, there is no doubt farmers would benefit from PLF. But when it comes to a specific technology, the applicable conditions and the urgency of application of a new technology need to be evaluated to obtain better application effect. PLF is a technical concept represents the development trend rather than a specific technology. The failure use of a specific technology in some cases can’t negate the positive effect of PLF itself.

Line 356-359: check this sentence, there are some mistakes and it’s very hard to read.

Author Response

We would like to thank the reviewers. Their valuable input has greatly contributed to the improvement of this paper.

In the following, we would like to address remarks made by the reviewers concerning content, scientific validity and completeness of the article. Recommendations merely concerning language, wording and grammar have been implemented in the text and are not further discussed here. The manuscript has once more been assessed for readability and changes have been made to improve it.

Reviewer 3:

Some of the recommendations concern language or structure of the text and have been addressed in the revised version. We would like to state specifically on the following remarks:

Line 41: The quotations have been checked. We understand that the reviewer points to the necessity of carefully interpreting headlines like the ones used. The sentences have been changed accordingly.

Line 53: In accordance with the recommendation of another reviewer, the definition has been changed.

Line 102: The reviewer is right -in pointing to “non-cow-sensors” being important as well. These sensors are repeatedly mentioned throughout the paper. The previous section referring to Knight (2020) has been adapted to make this clearer.

Line 141: It is not quite clear what the reviewer intended – The explanation of fig. 3 was slightly revised and the reference number added.

Line 135ff: The part has been expanded.

Line 143ff: A section on the demands to an algorithm used in PLF has been added, also providing additional references. The subject of “machine learning” is shortly described. We agree that “deep learning” will play an important role in the future, but its use is in our field mostly limited to the experimental and development stage and hence not to be covered in this paper.

Line 174: It is correct that analyses and trials are of utmost importance to gauge the reliability and usefulness of a certain algorithm or system. It is also true, however, that generally companies marketing sensor systems will not openly inform on the exact design of the algorithm. “Model” has been replaced by “algorithm”, this should avoid misunderstandings.

Line 266-267. We agree with the reviewer. A sentence stating the need to consider behavioural aspects has been added to conclude the section.

Line 277: There is hardly any disagreement on the positive impact PLF will have and already is having on livestock husbandry. However, most information stems from “testimonials”, anecdotal knowledge etc. The difficulty in assessing a positive impact has been described in the article. It is in accordance with other authors to state that relevant, evidence-based assessment of the presumed positive impact is yet to happen. As far as there is a documented impact from PLF in dairy farms, those articles have been included in our references e.g., Kempf (2016); Lora et al. (2020); Adenuga et al. (2020).

Round 2

Reviewer 2 Report

Paper has been improved from the first version. Some changes are required:

The reference no. 38 is not correctly. It cannot be actually referred to a unsupervised machine learning algorythm but to a supervised one.

Author Response

The reviewer remarks on the reference of Carslake et al. (2020), stating that the development described is not an unsupervised process. The reviewer is right as visual observations of behaviour have been fused with the sensor data . The satatement has been removed from the text.

Reviewer 3 Report

There was substantial improvement in the quality of the article. Make sure to proofread the manuscript again.

Author Response

The paper has oncemore been assessed for quality and correctness in writing.